# Cutaneous Hyalohyphomycosis and Its Atypical Clinical Presentations in Immunosuppressed Patients

**DOI:** 10.3390/life14010154

**Published:** 2024-01-21

**Authors:** Nikola Ferara, Sanja Špoljar, Liborija Lugović-Mihić, Ana Gverić Grginić, Violeta Rezo Vranješ, Iva Bešlić, Judita Perović, Tihana Regović Džombeta

**Affiliations:** 1Department of Dermatovenereology, Sestre Milosrdnice University Hospital Centre, 10000 Zagreb, Croatia; nikola.ferara@kbcsm.hr (N.F.); sanja.spoljar@kbcsm.hr (S.Š.); ivaaabukvic@gmail.com (I.B.); judita.perovic@gmail.com (J.P.); 2Department of Biology, Faculty of Science, University of Zagreb, 10000 Zagreb, Croatia; 3School of Dental Medicine, University of Zagreb, 10000 Zagreb, Croatia; 4Department of Microbiology, Parasitology and Hospital Infections, Sestre Milosrdnice University Hospital Centre, 10000 Zagreb, Croatia; ana.gveric-grginic@hzjz.hr; 5Department of Clinical and Molecular Microbiology, University Hospital Centre Zagreb, 10000 Zagreb, Croatia; violeta.rezo.vranjes@kbc-zagreb.hr; 6Clinical Department of Pathology and Cytology, Sestre Milosrdnice University Hospital Center, 10000 Zagreb, Croatia; tihana.dzombeta@kbcsm.hr; 7Department of Pathology, University of Zagreb School of Medicine, 10000 Zagreb, Croatia

**Keywords:** cutaneous fungal infections, cutaneous hyalohyphomycosis, invasive fungal infections, immunosuppressed patients, *Purpureocillium lilacinum*, *Fusarium*, *Acremonium*, mycetoma, spectrum of cutaneous infections

## Abstract

There has been a substantial increase in the number of cases of invasive fungal infections worldwide, which is associated with a growing number of immunosuppressed patients and a rise in antifungal resistance. Some fungi that were previously considered harmless to humans have become emerging pathogens. One of them is *Purpureocillium lilacinum*, a ubiquitous filamentous fungus commonly found in the environment, especially in the air and soil. *P. lilacinum* belongs to a bigger group of hyaline fungi that cause hyalohyphomycosis, a fungal infection caused by fungi with colorless hyphae. Although this is a heterogeneous group of fungi, there are similarities regarding their ubiquity, ways of transmission, affected patients, and difficulties in diagnostics and treatment. In hyalohyphomycosis, the skin is one of the most affected organs, which is why the involvement of dermatologists is crucial for the initial assessment, since the timely recognition and early diagnosis of this condition can prevent life-threatening infections and death. In this review, we covered cutaneous hyalohyphomycosis caused by *P. lilacinum* and other fungi in the same group, including *Fusarium*, *Penicilium*, *Scedosporium*, *Scopulariopsis*, *Acremonium*, and *Trichoderma* genera.

## 1. Introduction

There are approximately six million different fungal species worldwide, but less than 1% of them are known to infect humans. Fungi are typically part of the human skin microbiome, but if given certain circumstances, in immunosuppressed individuals, they can easily disseminate throughout the human body and cause severe infections or even death. According to some estimations, fungi cause around one million deaths per year worldwide. The most prominent and potentially severely pathogenic and invasive fungi are *Cryptococcus neoformans*, *Aspergillus* spp., and *Candida* spp. [1]. However, probably due to the growing population of immunosuppressed hosts and a rise in antifungal resistance, there have been increasing numbers of invasive infections caused by fungi that are not known for their pathogenic potential [2]. One of them is *Purpureocillium lilacinum* (formerly known as *Paecilomyces lilacinus*), a ubiquitous, saprophytic, asexual, and filamentous fungus found in the environment, especially in the soil, air, and water [3]. The infection caused by *P. lilacinum* is called hyalohyphomycosis, which is a broader term used to describe infections caused by different molds that present microscopically with hyaline (clear, light-colored, or colorless) septate hyphae, as opposed to other molds, which have dark-colored hyphae and cause an infection that is termed phaeohyphomycosis [4]. Because *P. lilacinum* is a growing pathogen in immunosuppressed individuals, it often causes cutaneous infections with elusive clinical presentations, and there is scarce information on its management [5], in this paper we performed a comprehensive review on cutaneous infections caused by this fungus. Additionally, we covered other important fungal pathogens that are known to cause cutaneous hyalohyphomycosis, including *Fusarium*, *Penicilium*, *Acremonium*, *Scopulariopsis*, and *Trichoderma* genera.

## 2. Hyalohyphomycosis

Hyalohyphomycosis is a broad term used to describe infections caused by colorless septate fungal hyphae in the affected tissue [6]. Therefore, hyalohyphomycosis does not represent a unique clinical syndrome and is primarily defined histologically. That is why fungi belonging to this group are extremely heterogeneous [4]. Hyalohyphomycosis commonly includes infections caused by species of *Fusarium*, *Scedosporium*, *Purpureocillium*, *Acremonium*, *Penicillium*, *Scopulariopsis*, *Trichoderma*, and others [7]. Table 1 contains basic information on the epidemiology, infection pathway, disease spectrum, and sensitivity to antifungals of the aforementioned fungi. In routine histologic sections, these fungi are often misidentified as *Aspergillus* spp. [6]. Because of this, and because of the fact that *Aspergilus* is far more common in isolates, fungi belonging to this group are often termed non-*Aspergilus* hyaline fungi. A definitive diagnosis of hyalohyphomycosis requires the isolation of the fungal organism, in the form of a positive fungal culture [6,8]. The exact identification of the fungus is paramount due to the often-high intrinsic resistance of many of the fungi in this group to most widely used antifungal agents (see Table 1) [9,10]. Hyalohyphomycosis is an emerging infection among immunosuppressed individuals, ranging from cutaneous forms to severe systemic and disseminated infections. Almost all of the fungi from this group are widely distributed in the environment and can enter the human body via inhalation or inoculation in a site of skin or mucosal breakdown [11,12,13]. Altogether, skin is the most frequently affected organ, either as the site of primary infection or as affected tissue in the disseminated disease, with variable clinical presentations [9,14,15,16]. Cutaneous hyalohyphomycosis ranges from superficial skin infections, including onychomycosis, to deep cutaneous and subcutaneous infections. Histopathologically, many of these fungi tend to invade blood vessels and produce hemorrhagic necrosis and tissue infarction [6], clinically presented as ulcerative lesions (e.g., *Purpureocillium lilacinum*) or eschars (e.g., *Fusarium* spp.) [17,18]. Figure 1 summarizes the spectrum of cutaneous hyalohyphomycosis, and associates various forms of cutaneous infections with the most common causative agent according to the reported cases. Because cutaneous infections can easily progress into disseminated ones, especially in immunosuppressed patients, and skin infections can sometimes be the first sign of a disseminated disease acquired via some other infection pathway besides direct inoculation at the site of a skin or mucosal breakdown (e.g., inhalation or ingestion), it is of extreme importance to diagnose cutaneous hyalohyphomycosis in a timely manner.

This figure associates various forms of cutaneous infections with the most common causative agent. The most dominant clinical presentations of cutaneous infections are subcutaneous and cutaneous nodules and ulcerative and necrotic lesions. Almost all of the hyaline fungi presented here can cause a disseminated disease with a variable clinical presentation. This figure additionally features three fungi that are not presented in the text—*Petriella setifera* [19], *Onychola canadensis* [20], and *Paraengyodntium album* [21]—but that have been identified as causative agents of hyalohyphomycosis, although not as frequently as the rest of the fungi.

**Table 1 life-14-00154-t001:** Summary of distribution, pathogenicity, infection pathways, possible infections in humans, and sensitivity to antifungal agents of the presented hyaline fungi. This table was made according to the available literature. More information on cutaneous infections is given in Figure 1. The sensitivity to antifungal agents presented here is a gross approximation—deviations may occur.

Fungus	Distribution and Pathogenicity	Infection Pathways	Infection Spectrum	Sensitivity to Antifungal Agents
*Purpureocillium* *lilacinum*	- Ubiquitous hyaline fungus - Widely distributed in environment, especially soil - Used as a biological nematocide [22]	- Inhalation - Direct inoculation at a site of skin or mucosal breakdown - Contaminated medical supplies, catheters, or prostheses [23]	- Mostly immunosuppressed individuals, but can cause skin or other infections in immunocompetent individuals - Most common infections sites: skin and subcutaneous tissue, eyes, sinuses, lungs, and central nervous system (CNS) [9] - Can cause disseminated infections and result in fatal outcome	Low: amphotericin B, fluconazole, flucytosine, and itraconazole Medium to high: posaconazole and voriconazole [5]
*Fusarium* spp.	- Genus of saprophytic filamentous fungi found in soil, in water systems, and on plants - Most common pathogenic species: *F. oxysporum*, *F. solani*, *F. moniliforme*, *F. chlamydosporum* [24], *F. proliferatum* [25], and *F. subglutinans* [26]	- Inhalation - Direct inoculation at a site of skin or mucosal breakdown - Contact with infected soil	- Infections in both immunosuppressed and immunocompetent individuals - Typically invasive in patients with deep or prolonged neutropenia and/or severe T cell immunodeficiency [10] - Superficial infections (skin, onychomycosis, and paronychia) - Eye infections (e.g., keratitis) - Deep cutaneous and subcutaneous infections - Disseminated infections affecting brain, bones, heart, and others	Low: itraconazole, isavuconazole, fluconazole, and echinocandins (micafungin, caspofungin, and anidulafungin) Medium: posaconazole and amphotericin B Species-dependent: voriconazole [27]
*Scedosporium* spp.	- Saprophytic fungus widely distributed in nature, mostly in soil, sewage, fertilizers, and rotten vegetation - *Scedosporium apiospermum* was formerly known as *Pseudoallerscheria boydii* [28]	- Inhalation - Ingestion - Direct inoculation at a site of skin or mucosal breakdown	- Causes mycetoma in both immunosuppressed and immunocompetent individuals [16] - Invasive infections in immunosuppressed individuals affecting lungs, bones, joints, and brain - Infections of central nervous system are often lethal due to delayed diagnostics and high antifungal resistance	Low: amphotericin B (intrinsically resistant) and most of the azoles Medium to high: voriconazole and micafungin + voriconazole [29]
*Penicilium* spp.	- Genus of more than 300 ubiquitous species found in soil, vegetation, air, and various food products - The most important pathogen is *Penicilium marneffei* (according to a genetic analysis, preferably called *Talaromyces marneffei*) - Non-*marneffei* species are reported to be pathogenic as well	- Inhalation of airborne conidia - Rare cases of direct inoculation at a site of skin breakdown for *P. marneffei*, and only one reported case for non-*marneffei* species [30]	- During the HIV/AIDS epidemic, *P. marneffei* was the third most common cause of opportunistic infections (after tuberculosis and cryptococcosis) in endemic regions of Southeast Asia and Southern China [12] - Today, it is a serious health threat to immunosuppressed travelers - Typical clinical presentation in immunosuppressed individuals: fever, lymphadenopathy, hepatosplenomegaly, and somewhat-typical cutaneous lesions	Low and variable: itraconazole Medium: amphotericin B and voriconazole High: terbinafine and echinocandins (caspofungin, micafungin, and anidulafungin) [31]
*Acremonium* spp.	- Genus of filamentous saprophytic molds widely found in nature, especially soil and decaying vegetation - These fungi have even been found in the rocks of continental Antarctica - The most common pathogenic species: *A. kiliense*, *A. egyptiacum* [32], and *A. strictum* [33]	- Inoculation at the site of a skin breakdown	- Mostly immunocompetent individuals following penetrating injury (e.g., walking barefoot) - Cutaneous and subcutaneous infections (onychomycosis and mycetoma) and eye infections (keratitis and endophthalmitis) - In immunosuppressed individuals, causes more severe cutaneous infections and systemic infections, such as pneumonia, arthritis, osteomyelitis, endocarditis, meningitis, and sepsis	Low: amphotericin B, flucytosine, and fluconazole High: terbinafine, posaconazole, and voriconazole [34]
*Scopulariopsis* spp.	- Saprophytic fungi found in soil, plant materials, food, air, and occasionally animals and humans - The most common pathogenic species: *S. brevicualis* [35]	- Local implantation or inoculation	- Keratinophilic species are predominantly associated with superficial infections of keratinized tissues (onychomycosis) - Can cause deep cutaneous lesions in severely immunosuppressed individuals	Low: itraconazole Intermediate: amphotericin B, voriconazole, and ketoconazole High: terbinafine and ciclopirox [36]
*Trichoderma* spp.	- Genus of fungi commonly found in soil - Used as a commercial biopesticide due to its effectiveness against soil-borne pathogens - Agricultural systems are assumed to be a major source of opportunistic mycoses [37] - The most common pathogenic species: *T. longibrachiatum*, *T. atroviride*, *T. bissetti*, *T. citrinoviride*, and *T. harzianum* [38]	- Direct inoculation at a site of skin or mucosal breakdown - Inhalation - Ingestion	- Broad spectrum of cutaneous infections (even necrotic ones) - Stomatitis - In invasive infections, it most commonly affects the lungs, peritoneum, and CNS [38]	Low: fluconazole Variable: amphotericin B and voriconazole High: echinocanidins (caspofungin) [38]

## 3. *Purpureocillium lilacinum*

*Purpureocillium lilacinum* is a ubiquitous hyaline fungus that is widely distributed in the environment. This fungus has a well-established place in agriculture as a biological nematicide, due to its ability to parasitize nematodes and their eggs while producing secondary metabolites that can promote plant growth [22]. Despite being previously considered an extremely rare pathogen in humans, it has the ability to cause infections of the skin and other sites in both immunosuppressed and healthy individuals [39].

### 3.1. Epidemiology

*P. lilacinum* is an emerging pathogen among immunocompromised individuals [23]. The most frequent predisposing factors for invasive infections are: hematological and oncological diseases (30.7% of invasive infections), solid organ transplantation (SOT), steroid treatments, and diabetes mellitus [5]. In this group, the most frequent type of infection is a local cutaneous one, followed by invasive sinusitis, pneumonia, and CVC (central venous catheter)-associated fungemia [2]. However, *P. lilacinum* also causes infections among immunocompetent individuals, mostly ocular infections (keratitis and endophthalmitis) related to ophthalmic surgery, non-surgical trauma, and skin infections [23]. It is also an opportunistic pathogen in infections associated with medical devices, such as a cardiac prosthesis or dialysis catheters [40]. A recent review identified 101 cases with invasive *P. lilacinum* worldwide in a period between 1974 and 2020, with the highest number of cases in the United States. Patients were mostly male (61.4%) and the median age was 53 years [5].

### 3.2. Pathogenesis

Due to the ubiquitous distribution of *P. lilacinum* in the environment, there are multiple possible modes of infection [41]. The most frequent infection sites are the skin, subcutaneous tissue, and eyes, although it can spread through the bloodstream, causing infections in various organs, such as the lungs, sinuses, and CNS [9]. Infection commonly occurs via the inhalation of the fungus and its consequent dissemination to the skin and other sites or directly, through inoculation at a site of skin or mucosal breakdown [11]. There have been reports of hospital-acquired *P. lilacinum* infections due to contaminated medical supplies, including lotions and catheters, or even tattoo-related infections, due to the contamination of the tattoo needle or ink [23]. An interesting feature of this fungus is that it can infect human phagocytic cells (macrophages and dendritic cells), thus escaping local immune defenses and migrating via the lymph flow [5]. Also, it exhibits the phenomenon of adventitious sporulation, which is the term used for the presence of fungal reproductive structures, phialides and conidia, within the infected tissue. This phenomenon is associated with a rapid rate of dissemination and a high prevalence of positive blood cultures, due to the sustained release of fungal spores into the bloodstream, along with angioinvasion, which has been observed in some other fungi, such as *Fusarium* spp. [42]. According to a recent review, *P. lilacinum* caused disseminated diseases in 22% of cases [5]. Both adventitious sporulation and the escape from the immune system could be responsible for the high rates of recurrent infections and the lack of a spontaneous resolution, as is often seen in *P. lilacinum* infections [40]. Although the immune response towards the fungus differs between immunocompetent and immunosuppressed individuals, *P. lilacinum* is capable of causing damage in both groups [18,43]. An experimental murine study showed that, despite not developing any clinical signs of infection, infected immunocompetent mice did have evident tissue damage, assessed using a histopathological analysis, which revealed conidia-like structures in the lung tissue of these mice [43]. Also, in contrast to previous studies, de Sequeira et al. found that *P. lilacinum* has the ability to infect and cause disease in immunocompetent and immunosuppressed mice with low levels of inoculum [43]. This could explain why *P. lilacinum* causes infections related to prostheses and medical devices.

### 3.3. Cutaneous Infections

The clinical presentations of cutaneous and subcutaneous *P. lilacinum* infections are variable, are non-specific, and can be misleading. They can vary from small erythematous papules and plaques with a central umbilication to hemorrhagic vesicles, soft or indurated cutaneous or subcutaneous nodules, or even cellulitis and ulcerations (Figure 2 and Figure 3). In one experimental murine study, the subcutaneous inoculation of *P. lilacinum* caused comparable damage to animal tissue, including dermatitis, panniculitis, and skin ulcerations with a diffuse inflammatory infiltrate in both immunosuppressed and immunocompetent mice. However, the lesions in immunosuppressed mice were more severe, including extensive areas of ulcers covered with crust, dermatitis, and suppurative panniculitis, with more fungal structures observed on histological slides [18]. Ulcerations are the result of angioinvasion, in both humans and mice. Even though some of the lesions can be dramatic and extensive, they are usually completely asymptomatic [39,41]. Skin infections are mostly located on the lower limbs, reinforcing the theory of skin being the inoculation site. Most skin infections are not accompanied by general symptoms, such as a fever or malaise [41]. The latest reports point out that *P. lilacinum* infections of the lower limbs in immunocompromised patient can be easily mistaken for typical bacterial cellulitis, caused by *Streptococcus* and *Staphylococcus*, and *P. lilacinum* is usually suspected only after the patient is unresponsive to antibiotic therapy [11]. A distinctive feature of *P. lilacinum* infections is the complete lack of a spontaneous resolution and a tendency towards recurrent infections [40], which is why they need to be properly diagnosed, taken seriously, and treated accordingly.

### 3.4. Diagnosis

The gold standard for making a diagnosis of a *P. lilacinum* infection is cultivation from lesions suspected to be the sites of infection [39]. *P. lilacinum* can grow on a conventional fungal culture medium in a rather rapid manner, and mature colonies can be obtained within three days [40]. Its growth is characterized by violaceous colonies with a woolly surface, while microscopic examination reveals branching, hyaline hyphae and phialides tapering at their distal end as chain-like conidiophores [39]. However, its growth in a culture can sometimes be the result of a contaminated laboratory environment. Moreover, due to its ubiquity, *P. lilacinum* is usually considered a contaminant in cultures until it is confirmed through a histopathological analysis. That is why it is mandatory to obtain a skin biopsy, since a histological examination with routine stains can detect hyphae and reproductive structures, such as phialides and conidia [40]. Additionally, the Grocott methenamine silver (GMS) stain is commonly used, since it imparts a black color to the fungal profiles and a pale green color to the background [44]. Due to its ability to sporulate in tissues, *P. lilacinum* can be confused with *Blastomyces dermatitidis*, but it is differentiated by the presence of hyphal elements within a tissue biopsy, elevated (1-3)-β-d-glucan, and growth on cultures [45]. In doubtful cases, it is advisable to confirm the diagnosis with MALDI-TOF mass spectrometry, DNA sequencing, or polymerase chain reaction (PCR) amplification of the 18 S RNA [40]. Identifying the exact fungus is of utmost importance due to the major differences in the sensitivity to antifungal agents within the same species, with an example being *Purpureocilium lilacinum* vs. *P. variotti* [41]. Additionally, when a disseminated disease is suspected, imaging techniques should be performed to assess the severity of the disease. The most commonly used imaging techniques are chest and paranasal CT (computed tomography) scans, followed by CNS imaging and the use of both a CT scan and MR [5]. Figure 4 summarizes the clinical presentation, the patient characteristics, and the diagnostic steps in a *P. lilacinum* infection.

### 3.5. Treatment

There is no standard treatment for a cutaneous *P. lilacinum* infection, and treatment is often difficult. Clinical management consists of an antifungal treatment, surgery, or a combination of both [46]. This fungus is intrinsically resistant to many antifungal agents, including itraconazole, terbinafine, griseofulvin, and amphotericin B [9,46]. Because of this, the treatment of *P. lilacinum* infections should be tailored according to the in vitro susceptibility results. Second-generation triazoles, such as voriconazole, posaconazole, isavuconazole, and ravuconazole, are promising treatment options [3]. Most of the recent reports show that posaconazole and variconazole have the lowest minimum inhibitory concentration (MIC) [5]. In most of the successful treatments of cutaneous *P. lilacinum* infections, voriconazole was the agent of choice [23]. It is also the preferable agent if the CNS is involved [3]. Accetta et al. reported the first successful treatment of an invasive *P. lilacinum* infection with isavuconazole [9]. Isavuconazole previously showed good results in treating invasive fungal infections in patients that were intolerant of variconazole and posaconazole [9]. It is interesting to note that combination therapy (e.g., amphotericin B + azole) did not result in a statistically significantly lower mortality rate in comparison to monotherapy (18.5% vs. 20%). Also, the use of amphotericin B is associated with a significantly higher mortality rate, which is in accordance with its intrinsic resistance [5]. The accumulated body of evidence shows that surgery plays an important part in the management of *P. lilacinum* infections (strength of recommendation: B, quality of evidence: III), and subcutaneous skin infections cure faster with surgery [46]. In a recent case report of a chronic subcutaneous infection of *P. lilacinum* in a female patient who received a hepato-renal allograft transplant, the authors emphasized the importance of a complete surgical intervention and foreign body search; complementary to antifungal agents, these interventions proved to be beneficial in preventing a recurrence or relapse of the cutaneous infection [41]. This was backed up by another case report in which a recurrent deep necrotic ulcer of the shin caused by *P. lilacinum* was successfully managed only after surgical debridement followed by split-thickness skin grafting, which resulted in the absence of recurrences at a two-year follow-up [39]. The necessity for surgical interventions in the treatment of *P. lilacinum* infections may be related to its ability to sporulate in tissues [3].

## 4. *Fusarium* spp.

### 4.1. Epidemiology and Pathogenesis

*Fusarium* is a genus of saprophytic filamentous fungi belonging to the Nectriaceae family, with a worldwide distribution [17]. *Fusarium* spp. can enter through airways or breaches in the skin and cause localized, locally invasive, or disseminated infections, which heavily depends on the immune status of the host [13]. Similar to *P. lilacinum*, *Fusarium* is also characterized by adventitious sporulation in the infected tissue [13], which is associated with recurrent infections and a tendency to disseminate [40]. The most common pathogenic species of this genus are *Fusarium oxysporum* and *Fusarium solani*, followed by *F. moniliforme* and *F. chlamydosporum* [24] (see Table 1).

### 4.2. Fusariosis

There are four types of fusariosis in humans: (1) superficial infections, including onychomycosis and paronychia; (2) keratitis and other eye infections; (3) deep localized infections; and (4) disseminated infections. The first two types of infections commonly occur in immunocompetent individuals, while the last two are seen in immunosuppressed individuals [47].

In immunocompetent individuals, cutaneous infections are usually localized and typically follow trauma or progress from onychomycosis [17]. Although *Fusarium* spp. are not typical causative agents of interdigital intertrigo or onychomycosis, they can cause these types of infections in immunocompetent individuals, most commonly *F. oxysporum* and *F. solani*. It has been proposed that these infections in immunocompetent individuals are the result of contact with soil infected with fungi, a high humidity, walking barefoot, and frequently visiting swimming pools [48]. In one clinical center in Thailand, *Fusarium* spp. were responsible for almost 10% of all nail and skin fungal infections [47]. There are also reports of *Fusarium* causing infection forms in immunocompetent individuals that highly resemble ecthyma gangrenosum (EG), which is generally considered pathognomonic for *Pseudomonas aeruginosa* or *Aeromonas* and is clinically defined as hemorrhagic pustules that lead to necrotic ulcers, which evolve into gangrenes with black scabs and, in later stages, become surrounded by a red halo [48]. Additionally, there is one report of a cutaneous *Fusarium* infection in an immunocompetent and healthy boy that presented as an asymptomatic verrucous plaque on the nose, without a previous history of trauma [49].

*Fusarium* may also act as a superinfecting agent of deep skin ulcers, third-degree burns, and surgical wounds [13,50]. Rajput et al. reported a localized *Fusarium* infection over a postsurgical scar that presented as a soft, nontender nodule, an abscess, and a discharging sinus, arranged linearly at the site of the operative scar [13].

In immunocompromised individuals, *Fusarium* spp. can cause a variety of invasive infections, including septic arthritis, endophthalmitis, osteomyelitis, sinusitis, keratitis, cystitis, brain abscesses, and disseminated infections [13,17,50]. Fusariosis is typically invasive, widespread, and potentially life-threatening in individuals with deep and prolonged neutropenia and/or severe T cell immunodeficiency [10]. An increased prevalence of invasive fusariosis has been noted in children with hematological malignancies who were treated with therapies that target cell surface antigens [51]. Although disseminated infections are mostly the result of the inhalation of the pathogen, cutaneous infections in immunocompromised individuals may also serve as the beginning point for dissemination [10]. On the other hand, almost 70–75% of patients with a disseminated infection have some form of cutaneous involvement [17,50], which is why dermatologists are needed to recognize and raise suspicion about disseminated diseases. There are various forms of clinical presentations of cutaneous *Fusarium* infections in a disseminated disease, including painful erythematous papules and nodules, gray-colored rounded lesions, multiple necrotizing skin lesions resembling ecthyma gangrenosum, targetoid lesions, subcutaneous nodules, and others [13,17,50]. Cutaneous lesions caused by *Fusarium* spp. tend to develop ulcerations and eschars due to thrombosis in the dermal vessels, the extravasation of erythrocytes, and focal dermal necrosis induced by fungal hyphae [46]. The lesions have a preference for the trunk and extremities, but can be found anywhere, including on the face, scalp, palms, and soles [50]. Cutaneous *Fusarium* infections are sometimes difficult to distinguish from infections caused by *Apergillus* or *Acremonium* spp., since the latter can also present as hemorrhagic or necrotic lesions and form eschars. However, a slight diagnostic clue is that *Aspergillus* tends to form larger (around 2–3 cm in size) and fewer lesions than *Fusarium* spp., which usually present with numerous small (around 1 cm in size) lesions [24].

### 4.3. Diagnosis and Treatment

A positive culture from obtained tissue remains the gold standard in diagnosing fusariosis. *Fusarium* grows in gray–white colonies on SDA (Sabouraud dextrose agar) and has characteristic sickle- or banana-shaped macroconidia [13]. PCR and serological tests are more sensitive methods for detecting *Fusarium* spp., but are more used in disseminated infections than in cutaneous ones [13]. A histopathological analysis of the infected tissue can help differentiate between *Fusarium* spp., which present with septate hyphae with branches at 45°, and *Aspergillus* spp., which present with branches at right angles [24]. One of the capital issues in the treatment of *Fusarium* infections is their relative resistance to most of the available antifungal drugs. There is a modest clinical response to amphotericin B, voriconazole, posaconazole, and some combination therapies [10].

## *5. Scedosporium* spp.

### 5.1. Epidemiology and Pathogenesis

*Scedosporium apiospermum*, formerly known as *Pseudoallescheria boydii*, is a saprophytic fungus widely distributed in nature, most commonly in soil, sewage, fertilizers, and rotten vegetation. It causes an opportunistic infection in severely immunosuppressed individuals, such as those with AIDS, lymphoma, or leukemia; SOT recipients; and others [52]. There are several modes of infection pathways, including inhalation, swallowing, and direct skin inoculation. The most frequent infection sites are the lungs, sinuses, bones, joints, eyes, and brain [52]. *Pseudoallescheria boydii*, a sexual form of *S. apiospermum*, is the most common cause of fungal pneumonia in cases of near drowning [53].

### 5.2. Cutaneous Infections

*S. apiospermum* is one of the most common causative agents of mycetoma (eumycetoma), a chronic, subcutaneous, granulomatous infection characterized by a triad of painless, subcutaneous, tumor-like swellings; multiple sinuses and fistulas; and discharged grains in pus. Similarly to other types of hyalohyphomycosis, mycetoma is also associated with a weakened immune system of the host, usually due to poor hygiene, malnutrition, or diabetes mellitus. It mainly affects the populations of the so-called “mycetoma belt”, which includes remote rural areas in tropical and subtropical countries [54]. Additionally, mycetoma can occur in immunocompetent individuals following trauma, even minor ones, mostly on the lower extremities [16]. Besides mycetoma, *S. apiospermum* can also cause cutaneous and subcutaneous granulomas [16]. Cutaneous lesions caused by *S. apiospermum* can gradually reach the muscles and the bones [55]. In immunosuppressed individuals, localized cutaneous infections can eventually progress, causing dissemination with a poor outcome [16].

### 5.3. Diagnosis and Treatment

A high mortality rate for disseminated diseases and CNS infections are mostly due to a delayed diagnosis [56], which is not surprising, taking into account the fact that, on average, 3.2 weeks are required to obtain a positive culture after the appearance of clinical symptoms [52]. *Scedosporium apiospermum* is known for its high antifungal resistance, which contributes to poor outcomes in disseminated infections. The most successful treatment options are azole antifungals, such as voriconazole or itraconazole [57], even though there is one report of a rather successful treatment with terbinafine, despite the relatively high in vitro MIC [55]. Just like in cutaneous *P. lilacinum* infections, a surgical intervention, consisting of the excision of the infected area, combined with antifungal medication proved to be beneficial in treating cutaneous *S. scedosporium* infections [55].

## *6. Penicilium* spp.

### 6.1. Epidemiology and Pathogenesis

*Penicilium* is a genus of fungi consisting of more than 300 ubiquitous species that can be found in soil, vegetation, air, and various food products [30]. The most important pathogen from this genus is *P. marneffei*; today, from a generic point of view, it is preferably called *Talaromyces marneffei* [15], but for the sake of simplicity and tradition, we present it in this genus. *Penicilium marneffei* is a dimorphic fungus that can cause life-threatening infections in immunosuppressed individuals, especially patients with HIV/AIDS [8]. During the HIV/AIDS epidemic, it was known as the third most common cause of opportunistic infections, after tuberculosis and cryptococcosis, in the endemic regions of Southeast Asia and Southern China [12]. Today, due to effective prophylaxis in HIV+ patients, *P. marneffei* has become more important as a serious health threat to immunosuppressed travelers, rather than an opportunistic agent in the HIV group. The most prevalent infection pathway is the inhalation of airborne conidia, although there are rare reports of infections after direct skin inoculation [12]. There is also evidence of seasonality in *P. marneffei* infections, with increased cases noted during the rainy months [58].

### 6.2. Cutaneous Infections—Presentation, Diagnosis, and Treatment

The typical clinical presentation in immunosuppressed individuals includes a fever, lymphadenopathy, hepatosplenomegaly, and cutaneous lesions. The cutaneous lesions caused by *P. marneffei* are rather typical, and in around 70% of infected patients, they appear as centrally umbilicated papules resembling *Molluscum contagiousum*, nodules and necrotic lesions, mostly located on the face and upper trunk [8,15]. Multiple abscesses can often appear [59]. Initially, simple papules ulcerate over time and change with the appearance of central necrotic umbilication. Unlike *Molluscum contagiousum*, which resolves by itself in six months to two years in HIV patients, *P. marneffei* lesions lack spontaneous resolution and require an active antifungal treatment [58]. Dermoscopy can be useful for differentiation, since the umbilicated papules in *P. marneffei* infections have a round, whitish, amorphous structure as the most common finding [60]. Clinically and histopathologically, *P. marneffei* resembles disseminated histoplasmosis, since both proliferate within histiocytes and produce lesions of a similar size (2–5 mm) [4]. Culture cultivation is essential for diagnosing infections, and the antifungal agents of choice are amphotericin B, voriconazole, and fluconazole, with variable clinical responses [8]. Terbinafine and echinocandins are highly active in vitro against *Penicilium* spp. (see Table 1), but these antifungal agents are not widely used for treating invasive infections due to these fungi [31].

### 6.3. Non-Marneffei Species—Trends

Also, non-marneffei species are increasingly recognized as a rare cause of invasive infections in patients with hematologic malignancies, with the highest incidence in patients with acute leukemia. There has been a recent report on a cutaneous infection caused by *Penicilium cluniae* in a patient with acute myelogenous leukemia that presented as a shin nodule with a central eschar [30], which is likely the first described case of a localized cutaneous infection due to this fungus, since all the other cases have reported pulmonary or disseminated infections. This points to inoculation as an infection pathway, which could mean that various other fungi belonging to this genus are capable of causing localized cutaneous infections.

## *7. Acremonium* spp.

### 7.1. Epidemiology and Pathogenesis

The *Acremonium* genus consists of filamentous saprophytic molds that can be widely found in nature, especially in soil and decaying vegetation [61,62]. Their ubiquity is underlined by the fact that *Acremonium* isolates are found even in the rocks on continental Antarctica [32]. Therefore, it is not surprising that these molds are among the common laboratory contaminants [61]. Although many species belonging to this genus are speculated to cause infections in humans, recent data suggest that *A. kiliense* and *A.egyptiacum* are the two species to which most of the infections can be attributed [32]. The authors of the relevant research emphasize the doubtfulness of fungal identification (e.g., *A. strictum* or *A. egyptiacum*) in many of the reported cases of infections in humans, since it heavily relies on morphological criteria [32].

### 7.2. Cutaneous and Other Infections

*Acremonium* spp. can cause cutaneous and subcutaneous infections, such as superficial skin infections, onychomycosis, mycetoma, keratitis, and endophthalmitis in lens users and following trauma and systemic infections, including pneumonia, arthritis, osteomyelitis, endocarditis, meningitis, and sepsis, in immunocompromised individuals [33,61]. The introduction of the fungus through a penetrating injury often leads to localized infections in immunocompetent individuals [33]. That is why mycetoma can be seen in developing countries in individuals who walk barefoot [62]. In immunosuppressed individuals, skin infections are more severe and can include pustules and purulent exudate, ulcerations, painless swelling, necrotic areas, nodules, and fistulae. According to a review, cutaneous and subcutaneous infections have occurred on different body parts—the face, cheeks, upper legs, hands, legs, ankles, and feet [32].

### 7.3. Diagnosis and Treatment

As with other fungi, a positive culture is essential for the diagnosis of an infection caused by *Acremonium* spp. However, it is difficult to accurately identify the members of *Acremonium* spp. in routine clinical microbiology laboratories, since they typically grow slowly—taking up to 14 days—and the exact identification of the species usually requires molecular diagnostics [63]. Since *Acremonium* is one of the common laboratory contaminants, a biopsy and a subsequent histopathological analysis are obligatory to confirm the infection. Sometimes, a histopathologic examination can be misleading because the hyphae and the branching can resemble other agents, such as *Aspergillus* spp. [63]. *Acremonium* spp. have a variable susceptibility to antifungal agents, but there are reports of a high sensitivity to terbinafine, posaconazole, and voriconazole and a rather low sensitivity and high MIC for amphotericin B, flucytosine, and fluconazole [34] (Table 1). Interestingly, similar to *Fusarium* and *P. lilacinum*, *Acremonium* has the feature of adventitious sporulation with the sustained release of fungal spores into the bloodstream [4], which is associated with a high rate of positive blood cultures and recurrent infections. That is why, whenever possible, a surgical treatment of cutaneous and subcutaneous lesions should be performed, just like in the case of *P. lilacinum* infections [64].

## 8. *Scopulariopsis* spp.

### 8.1. Epidemiology

*Scapulariopsis* spp. are saprophytic fungi found in soil, plant materials, food, air, and occasionally, animals and humans [35,65]. *Scopulariopsis brevicaulis* is the most common species and the only one that has been identified in human infections by using DNA sequencing [35]. In humans, *Scapulariopsis* spp. are mainly associated with superficial infections of keratinized tissues, but they are also causative agents of cutaneous, subcutaneous, and deep-tissue infections and disseminated infections [66].

### 8.2. Onychomycosis

Since *Scopulariopsis* spp. are known to be keratinophilic, *S. bravicaulis* has proven to be one of the predominant species among non-dermatophytic filamentous fungi in onychomycoses [66]. A retrospective analysis in the Croatian population showed that *S. brevicaulis* was isolated from nail, skin, and scalp scrapings, and most of the patients from whom the specimens were obtained lived in rural settings, working as farmers in close contact with the soil and domestic animals. Additional predisposing factors for infections were pre-existing dermatoses (atopic dermatitis or psoriasis), lower-extremity circulatory insufficiency, trauma, and metabolic disorders [65]. In many cases, it is difficult to distinguish onychomycosis caused by *Scopulariopsis* spp. from the dermatophytic one, but there are two slight clinical clues for a *Scopulariopsis* infection of the toenail—the absence of tinea pedis (a dermatophyte) and the fact that the other toenails are usually unaffected [4].

### 8.3. Other Infections

Onychomycosis can easily progress into an invasive infection in immunosuppressed individuals. In the case report of a neutropenic patient, a *Scopulariopsis* infection presented as a painless purpuric cutaneous lesion at the top of the toe along with skin hyperkeratosis near the nail [67]. Deep cutaneous infections have been described in various anatomical sites and with a broad clinical presentation, such as erythematous papules and plaques, nodules, tumors, ulcerations, swellings, and even necrosis [35]. In a disseminated infection, *Scapulariopsis* can present as multiple dark skin lesions on the trunk, as reported in a patient with diffuse large B cell lymphoma, treated with chemotherapy and an autologous stem cell transplant [68]. There has also been a report of keratitis caused by *Scopulariopsis* following LASIK (laser in situ keratomileusis) [69].

### 8.4. Diagnosis and Treatment

Unlike dermatophytic infections, which are diagnosed by isolating the dermatophytes from the site of infection, the isolation of *Scapulariopsis* does not always indicate an infection, mainly because of its ubiquity. Therefore, in diagnosing onychomycosis, a KOH examination and culture isolation are performed [70]. Although the isolation of *Scopulariopsis* spp. is rather easy due to their growth on routine laboratory media, it may be difficult to identify the exact species of fungus. That is why, in a setting of an invasive or disseminated infection, molecular methods such as PCR, DNA sequencing, or probe hybridization are more frequently used [71].

*Scapulariopsis* is also inherently resistant to most antifungals. In a study by Skóra et al., terbinafine and ciclopirox exhibited the best antifungal activity against *S. brevicaulis*, while itraconazole, ketoconazole, and voriconazole proved to be the least effective, exhibiting the highest MICs [36]. However, there were some promising results with isavuconazole (MIC of 2 μg/mL) and combination therapies, such as caspofungin + posaconazole + terbinafine [35]. Also, the successful treatment of an invasive *Scopulariopsis* infection usually requires aggressive surgical debridement [4].

## *9. Trichoderma* spp.

### 9.1. Epidemiology

*Trichoderma* is a genus of fungi whose members can cause invasive fungal infections in immunosuppressed individuals, mainly in those with hematological malignancies and in peritoneal dialysis patients. *Trichoderma longibrachiatum* is the most commonly reported species in invasive fungal infections, followed by other species such as *Trichoderma atroviride*, *Trichoderma bissettii*, *Trichoderma citrinoviride*, *Trichoderma harzianum*, *Trichoderma koningii*, *Trichoderma pseudokoningii*, and *Trichoderma viride* [38]. Up until recently, these fungi were considered harmless colonizers, and just like other fungi that cause hyalohyphomycoses, they are commonly found in the environment, particularly in soil [72]. Moreover, *Trichoderma* is effective against common soil-borne pathogens and is, therefore, used as a commercial biopesticide [38]. According to some investigators, its mentioned use in agricultural systems could be a major source of emerging human mycoses caused by *Trichoderma* [37].

### 9.2. Cutaneous and Other Infections

The proposed infection pathways are either via a skin or mucosal breakdown or the inhalation or ingestion of the fungus [4]. In invasive infections, the most frequently affected organs are the lungs, peritoneum, and CNS [38]. There are several reports of cutaneous lesions forming at the site of a peripheral intravenous cannula or central venous catheter, presenting as ulcero-necrotic lesions and erythematous-indurated, centrally necrotic plaques [4]. In one of the described cases of a disseminated disease, a *Trichoderma* infection presented as small pustules and indurated plaques on the skin of the extremities, at some sites, especially around an intravenous cannula, progressing into necrosis resembling ecthyma gangrenosum [72]. Cutaneous infections caused by *Trichoderma* often present with ulcers and necrosis and can be similar to cutaneous *Fusarium* infections, which is why *Trichoderma* spp. infection is one of the major diffferential diagnosis of cutaneous fusariosis [72]. There is also a report of fatal necrotizing stomatitis caused by *Trichoderma longibrchiatum* in a neutropenic patient with malignant lymphoma, which presented as a wide, destructive, gingival ulcer covered by a violaceous necrotic pseudo-membrane [73].

### 9.3. Diagnosis and Treatment

A positive culture is essential for a definitive diagnosis, and *Trichoderma* is known for its fast growth [4], developing initially smooth or translucent, and later, floccose, colonies with white and green rings [6]. Invasive *Trichoderma* infections are mostly associated with a poor outcome. In treating cutaneous lesions, it is advisable to perform a surgical excision of the infected tissue, followed by antifungal therapy. There have been reports of the successful treatment of invasive infections by using a combination therapy of voriconazole and caspofungin [38].

## 10. Differential Diagnoses of Cutaneous Hyalohyphomycosis

Immunosuppression should always raise the suspicion of fungal involvement, especially if there is an atypical clinical presentation. Additionally, antifungal prophylaxis is often unsuccessful due to frequent adverse events and a poor response related to antifungal resistance [74]. Due to the heterogeneous clinical presentation, there are numerous differential diagnoses of cutaneous hyalohyphomycosis. Mycetoma and cutaneous and subcutaneous nodules should be differentiated from both infectious and non-infectious diseases, including tuberculosis, fibroma, rheumatoid nodules, keloids, fibrolipoma, dermatofibroma, dermatofibrosarcoma protuberans, Kaposi’s sarcoma, malignant melanoma, and verrucous carcinoma [54]. When assessing nodules in hyalohyphomycosis, there are usually more than one. Dermoscopy is useful for differentiating solitary nodules from melanoma or keratinocyte carcinoma. Mycotic nodules predominantly do not have any alarming dermoscopic patterns. In most cases, a histopathological analysis of the biopsy or the whole lesion, along with the use of specific stains for the detection of fungal structures, such as periodic acid–Schiff (PAS), Alcian blue, safranin O, and GMS, resolves diagnostic doubt and can steer the diagnostics toward the exact identification of the causative fungus [44]. Ecthyma gangrenosum-like lesions caused by *Fusarium* spp. and *Acremonium* spp. need to be distinguished from the typical causative agents—*Pseudomonas aeruginosa* or *Aeromonas* spp. [75]. Cutaneous lesions with central umbilication, as typically seen in disseminated *P.marneffei* [15] or *P. lilacinum* [18], should be distinguished from Molluscum contagiosum. As already said, molluscum resolves by itself, while these fungi require active treatment [58].

Fungal cellulitis of the lower extremities can be mistaken for bacterial cellulitis and should be suspected if there is no response to antibiotic therapy. Additionally, open wounds often serve as an entry for bacteria, but less commonly do the wounds develop as a result of infection (with some exceptions); on the contrary, in fungal infections, ulcerations are more common and are the result of an invasion of the vascular supply [11]. Pustular lesions should be differentiated from impetigo, folliculitis, generalized pustular psoriasis, acute generalized exanthematous pustulosis (AGEP), and others [76]. Ulcerations occurring on atypical anatomical sites, such as the trunk or upper extremities, or those that cannot be associated with circulatory issues or other underlying conditions, should be evaluated for the fungal causative agent. Disseminated ulcero-necrotic lesions, as seen in infections caused by *Fusarium* spp. and *Trichoderma* spp. [4], could be mistaken for pityriasis lichenoides et varioliformis acuta (PLEVA), Behçet’s syndrome, pyoderma gangrenosum, embolic phenomena, calciphylaxis, antiphospholipid syndrome [77], or others.

Eschars should be evaluated for cutaneous anthrax, staphylococcal skin sepsis, cat-scratch disease, or a zoonotic viral infection [78]. Due to the possibility of a similar clinical presentation in lesions caused by bacterial and/or fungal pathogens, it is of the utmost importance to take samples for bacteriology and mycology analyses and to emphasize in the referral documentation for the microbiology laboratory that a fungal infection is suspected. Additionally, in the case of a suspected rare fungal infection, communication with the clinical microbiologist and pathologist is obligatory.

## 11. Conclusions

Due to a growing population of immunosuppressed hosts and a rise in antifungal resistance, there has been an increase in the number of invasive infections caused by fungi that are not known for their pathogenic potential. One of them is *P. lilacinum*, a ubiquitous, saprophytic, asexual, and filamentous fungus found in the environment, especially in the soil, air, and water. It belongs to a broader group of fungi that can cause hyalohyphomycosis, which is a term used for infections caused by fungi with colorless hyphae, also including the following genera: *Fusarium*, *Penicilium*, *Scedosporium*, *Scopulariopsis*, *Acremonium*, and *Trichoderma*. Even though the fungi belonging to this group are extremely heterogeneous, they share similarities in terms of their ubiquity, disease pathway, affected patients, and difficulties in diagnostics and treatment. Almost all of them are widely distributed in the environment and can enter the human body via inhalation or direct inoculation at a site of mucosal or skin breakdown. They can cause a variety of infections, ranging from superficial cutaneous infections to systemic infections affecting multiple vital organs, such as the lungs, brain, or heart, and disseminated infections with life-threatening potential. The severity of the infection depends on the immune status of the individual, and in the example of *P. lilacinum*, the conditions most frequently associated with invasive hyalohyphomycosis are: oncological or hematological diseases, a solid organ transplant, systemic steroid treatment, and diabetes mellitus. Nevertheless, in infections caused by *P. lilacinum* and most of the other fungi in this group, the skin is the most affected organ, either as the site of primary infection or in the disseminated disease, which is why the involvement of dermatologists is crucial for the initial assessment. Cutaneous hyalohyphomycosis is not a unique clinical syndrome, nor is it easy to recognize. It presents with a broad spectrum of lesions, including onychomycosis, erythematous papules and plaques, pustules, cutaneous and subcutaneous nodules, mycetoma, ulcerations, necrotic lesions, eschars, and others. Most of the fungi belonging to this group are capable of causing ulcerative and necrotic lesions, which is the result of an invasion of the vascular supply. Consequentially, necrotic lesions and eschars can occur, especially due to *Fusarium* spp., *Acremonium* spp., and *P. lilacinum* infections. In a disseminated disease, the clinical presentation is variable, but *P. marneffei* tends to develop a relatively typical presentation consisting of centrally umbilicated papules resembling *Molluscum contagiosum*. Some of the fungi, including *Fusarium* spp., *Acremonium* spp., and *P. lilacinum*, are known for their adventitious sporulation or the presence of reproductive elements in the infected tissue, which enables them to continuously release spores into the bloodstream, and this is associated with recurrent infections and a high frequency of positive blood cultures. In assessing cutaneous hyalohyphomycosis, it is important to consider many differential diagnoses. A positive fungal culture remains the golden standard for the diagnosis of this condition. However, since most of the fungi of this group are considered laboratory contaminants, a skin biopsy is crucial for the diagnosis, since it can detect fungal elements in the tissue. Additionally, the precise determination of the causative fungus with MALDI-TOF mass spectrometry, DNA sequencing, or PCR amplification of the 18 S RNA is of extreme importance because of the intrinsic resistance of the fungi to widely used antifungal agents. Treatment is difficult, due to the resistance to antifungal agents and a tendency for recurrent infections. If possible, it is advisable to perform a surgical excision of the infected tissue combined with systemic antifungal drugs. Since there is a trend of a growing immunosuppressed population worldwide, it is likely that cutaneous hyalohyphomycoses are going to be even more frequent in the future. Therefore, it is important for dermatologists to timely suspect the diagnosis and collaborate with microbiologists and pathologists in multidisciplinary team, to diagnose and treat this potentially life-threatening condition.

## Figures and Tables

**Figure 1 life-14-00154-f001:**
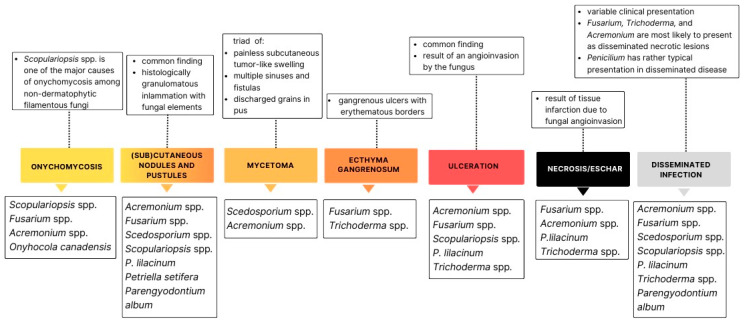
Summary of cutaneous lesions reported in hyalohyphomycosis.

**Figure 2 life-14-00154-f002:**
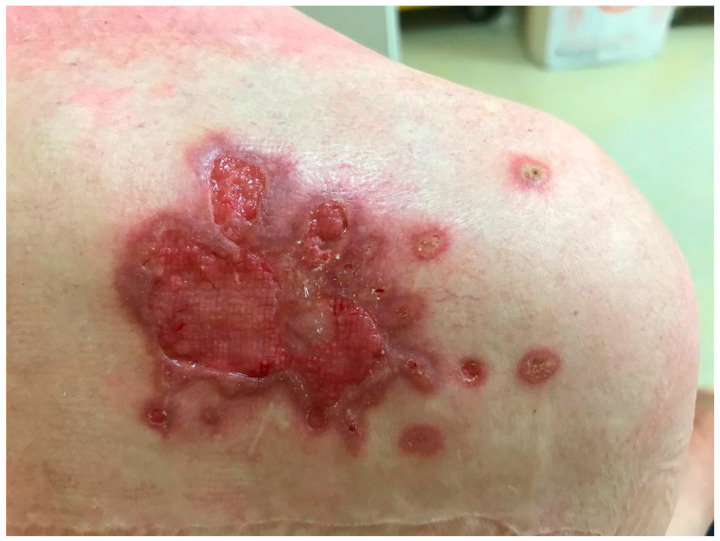
Ulcerative skin lesion on the skin of right scapula as a manifestation of *P. lilacinum* skin infection (from archives of the authors).

**Figure 3 life-14-00154-f003:**
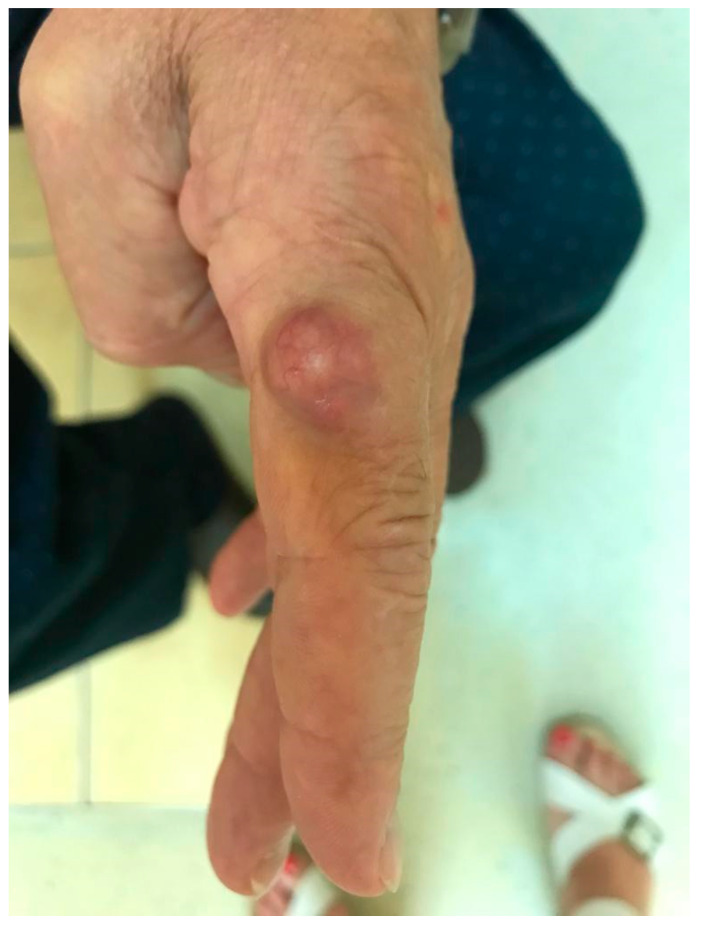
Cutaneous nodule on the left index finger as a manifestation of *P. lilacinum* infection (from archives of the authors).

**Figure 4 life-14-00154-f004:**
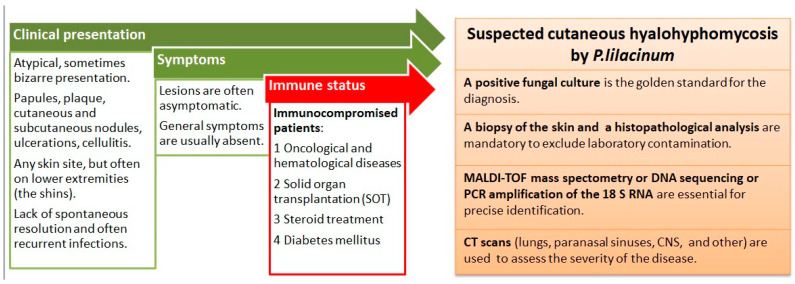
Summary of clinical presentation, patient characteristics, and diagnostic steps in *P. lilacinum* infection.

## Data Availability

No new data were created or analyzed in this study. Data sharing is not applicable to this article.

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
