# Peer review of "Cutaneous Hyalohyphomycosis and Its Atypical Clinical Presentations in Immunosuppressed Patients"

_life, 2024, doi:10.3390/life14010154_

Round 1
Reviewer 1 Report
Comments and Suggestions for Authors
Dear Authors,
Congratulations for your work. Below, you can find my comments and remarks regarding your manuscript:
Lines 64-66: The sentence “Since P. lilacinum is a growing pathogen in immunosuppressed individuals, often causes cutaneous infections with elusive clinical presentations and there is scarce information on its management” needs to be rephrased. Consider removing “since” from the beginning of the sentence or other corresponding changes.
Lines 80-81: You included both Paecilomyces and Purpureocillium in the same sentence, but since they are referring to the same cases, please consider mentioning only the current name for these pathogens.
Line 83: The word “aforementioned” is far more relevant than “mentioned”.
Lines 104-105: The part of the sentence “by another manner (than inoculation), it is of extreme importance to timely suspect cutaneous hyalohyphomycosis” needs revising to become more understandable for the readership.
Figure 1: Under the box corresponding to “(sub)cutaneous nodules and pustules,” the Scedosporium spp is mentioned twice. Please correct. In addition, you can add Petriella setifera below this category, which has been demonstrated to cause such lesions in the literature: Cerne, C., Seyedmousavi, S., & Bennett, J. E. (2021). Cutaneous hyalohyphomycosis due to Petriella setifera following traumatic inoculation in an immunocompetent host. Medical mycology case reports, 32, 56–60. https://doi.org/10.1016/j.mmcr.2021.03.002
// Under the “disseminated infection” box, Scedosporium spp is also mentioned twice. Please correct. // Skin infection due to Onychocola canadensis is not mentioned in Figure 1. A representative publication is Erbagci, Z., Balci, I., Erkiliç, S., Zer, Y., & Inci, R. (2002). Cutaneous hyalohyphomycosis and onychomycosis caused by Onychocola canadensis: report of the first case from Turkey. The Journal of dermatology, 29(8), 522–528. https://doi.org/10.1111/j.1346-8138.2002.tb00320.x // Skin infections due to Parengyodontium album are also not mentioned. A representative publication is Tsang, C. C., Chan, J. F. W., Pong, W. M., Chen, J. H. K., Ngan, A. H. Y., Cheung, M., Lai, C. K. C., Tsang, D. N. C., Lau, S. K. P., & Woo, P. C. Y. (2016). Cutaneous hyalohyphomycosis due to Parengyodontium album gen. et comb. nov. Medical mycology, 54(7), 699–713. https://doi.org/10.1093/mmy/myw025
Table 1: The correct spelling is “Purpureocillium lilacinum” and not “Purpucecillium lilacinum”. // Pathogenic species belonging to Fusarium spp and causing skin infections also include Fusarium proliferatum (Suzuki, H., Hashimoto, T., Sugiura, R., Ogata, H., Noguchi, H., Hiruma, M., Yaguchi, T., & Satoh, T. (2023). Disseminated cutaneous hyalohyphomycosis caused by Fusarium proliferatum in a patient with aplastic anemia. The Journal of dermatology, 50(6), e183–e184. https://doi.org/10.1111/1346-8138.16708), and Fusarium subglutinans (Chen, Q. X., Li, C. X., Huang, W. M., Shi, J. Q., & Li, S. F. (2010). Cutaneous hyalohyphomycosis caused by Fusarium subglutinans. European journal of dermatology : EJD, 20(4), 526–527. https://doi.org/10.1684/ejd.2010.0982). A relevant publication with great impact on the scientific community could offer in this section more insights (on Fusarium solani and oxysporum): Nucci, M., & Anaissie, E. (2002). Cutaneous infection by Fusarium species in healthy and immunocompromised hosts: implications for diagnosis and management. Clinical infectious diseases : an official publication of the Infectious Diseases Society of America, 35(8), 909–920. https://doi.org/10.1086/342328 . // Scedosporium apiospermum needs to be highlighted in the corresponding section. A representative publication is: Makino, K., Fukushima, S., Maruo, K., Egawa, K., Nishimoto, K., & Ihn, H. (2011). Cutaneous hyalohyphomycosis by Scedosporium apiospermum in an immunocompromised patient. Mycoses, 54(3), 259–261. https://doi.org/10.1111/j.1439-0507.2009.01797.x // Acremonium strictum is also a causative agent of cutaneous hyalohyphomycosis (though rare): Sharma, A., Hazarika, N. K., Barua, P., Shivaprakash, M. R., & Chakrabarti, A. (2013). Acremonium strictum: Report of a Rare Emerging Agent of Cutaneous Hyalohyphomycosis with Review of Literatures. Mycopathologia, Advance online publication. https://doi.org/10.1007/s11046-013-9709-1 // There is a reference in this Table, but in general, citations are lacking here. It is understandable not to cite publications for a Figure (when more details can be extracted from text), but you need to add more citations (even at the end of Table 1 legend altogether). The reference [40] in the Table does not agree the number of citations in the text.
Lines 132-135: Please rephrase the sentence, as it is difficult to read.
Line 142: CVC probably refers to central venous catheters, but it has to be explained in the text as has not been mentioned before.
Line 155: CNS acronym is first used in Table 1. Please make the necessary corrections.
Figure 4: Please correct the typo in “cellulitis”
Line 240: In the section “Treatment” a significant publication is missing, which includes a summary of recommendations for the treatment of Purpureocillium lilacinum infections: Tortorano, A. M., Richardson, M., Roilides, E., van Diepeningen, A., Caira, M., Munoz, P., Johnson, E., Meletiadis, J., Pana, Z. D., Lackner, M., Verweij, P., Freiberger, T., Cornely, O. A., Arikan-Akdagli, S., Dannaoui, E., Groll, A. H., Lagrou, K., Chakrabarti, A., Lanternier, F., Pagano, L., … European Confederation of Medical Mycology (2014). ESCMID and ECMM joint guidelines on diagnosis and management of hyalohyphomycosis: Fusarium spp., Scedosporium spp. and others. Clinical microbiology and infection : the official publication of the European Society of Clinical Microbiology and Infectious Diseases, 20 Suppl 3, 27–46. https://doi.org/10.1111/1469-0691.12465
Line 272: An increased prevalence of invasive fusariosis has been noted in children with hematological malignancies who were treated with therapies that target cell surface antigens: Kyriakidis, I., Vasileiou, E., Rossig, C., Roilides, E., Groll, A. H., & Tragiannidis, A. (2021). Invasive Fungal Diseases in Children with Hematological Malignancies Treated with Therapies That Target Cell Surface Antigens: Monoclonal Antibodies, Immune Checkpoint Inhibitors and CAR T-Cell Therapies. Journal of fungi (Basel, Switzerland), 7(3), 186. https://doi.org/10.3390/jof7030186 .
Line 552: Typo of “micology”
Line 555: Immunosuppression raises the suspicion of fungal involvement and it should be stated clearly in the differential diagnosis section. In one sentence, please refer to the reason for the failure of fungal prophylaxis in these cases. Moreover, host defenses against fungi rely on the sophisticated interplay between mucocutaneous barrier integrity (among others).
General comment: The separate sections for each hyalohyphomycosis agent can be shortened, as the text is wordy and complicated to read. You should focus only on cutaneous involvement and cite treatment guidelines from well-respected sources and committees (and not case reports).
Thank you.
Comments on the Quality of English LanguageComments on the quality of the English language have been incorporated in the review report above entitled "Comments and Suggestions for Authors
".
Author Response
Dear editor and reviewers,
We are thankful for your positive opinion about our paper and much needed expert suggestions! After implementing all of the corrections, we can say that this paper is now substantially better than the originally submitted version. I hope you are going to recognize our effort in trying to meet your corrective intentions. Please find corrected version of the manuscript in which we highlighted the corrected parts with comments of the reviewers so that it can be easily seen to what comment of the reviewer does the correction refer to. In the following we are going to specify our corrections (R1=reviewer 1, R2= reviewer 2).
R1: Lines 64-66: The sentence “Since P. lilacinum is a growing pathogen in immunosuppressed individuals, often causes cutaneous infections with elusive clinical presentations and there is scarce information on its management” needs to be rephrased. Consider removing “since” from the beginning of the sentence or other corresponding changes.
- We replaced “since” with “because” for better understanding. Thank you!
R1: Lines 80-81: You included both Paecilomyces and Purpureocillium in the same sentence, but since they are referring to the same cases, please consider mentioning only the current name for these pathogens.
- We excluded Paecilomyces from this sentence to avoid confusion. Thank you!
R1: Line 83: The word “aforementioned” is far more relevant than “mentioned”.
- Corrected!
R1: Lines 104-105: The part of the sentence “by another manner (than inoculation), it is of extreme importance to timely suspect cutaneous hyalohyphomycosis” needs revising to become more understandable for the readership.
- We revised the sentence for better understanding.
R1: Figure 1: Under the box corresponding to “(sub)cutaneous nodules and pustules,” the Scedosporium spp is mentioned twice. Please correct. In addition, you can add Petriella setifera below this category, which has been demonstrated to cause such lesions in the literature: Cerne, C., Seyedmousavi, S., & Bennett, J. E. (2021). Cutaneous hyalohyphomycosis due to Petriella setifera following traumatic inoculation in an immunocompetent host. Medical mycology case reports, 32, 56–60. https://doi.org/10.1016/j.mmcr.2021.03.002 // Under the “disseminated infection” box, Scedosporium spp is also mentioned twice. Please correct. // Skin infection due to Onychocola canadensis is not mentioned in Figure 1. A representative publication is Erbagci, Z., Balci, I., Erkiliç, S., Zer, Y., & Inci, R. (2002). Cutaneous hyalohyphomycosis and onychomycosis caused by Onychocola canadensis: report of the first case from Turkey. The Journal of dermatology, 29(8), 522–528. https://doi.org/10.1111/j.1346-8138.2002.tb00320.x // Skin infections due to Parengyodontium album are also not mentioned. A representative publication is Tsang, C. C., Chan, J. F. W., Pong, W. M., Chen, J. H. K., Ngan, A. H. Y., Cheung, M., Lai, C. K. C., Tsang, D. N. C., Lau, S. K. P., & Woo, P. C. Y. (2016). Cutaneous hyalohyphomycosis due to Parengyodontium album gen. et comb. nov. Medical mycology, 54(7), 699–713. https://doi.org/10.1093/mmy/myw025
- We removed Scedosporium spp which was mentioned twice in two places. We also added the suggested three fungal agents in the figure – Petriella setifera, Onychola canadensis and Parengyodontium album that can cause cutaneous hyalohyphomycosis, but are not that frequent. In the caption of the figure 1 we also clearly stated the inclusion of additional three fungal agents that were not mentioned in the text. We also made some other changes in the caption to correspond to the changes in the text, such as replacing word “eflorescence” with “forms of cutaneous infections” to avoid confusion in readers.
R1: Table 1: The correct spelling is “Purpureocillium lilacinum” and not “Purpucecillium lilacinum”. // Pathogenic species belonging to Fusarium spp and causing skin infections also include Fusarium proliferatum (Suzuki, H., Hashimoto, T., Sugiura, R., Ogata, H., Noguchi, H., Hiruma, M., Yaguchi, T., & Satoh, T. (2023). Disseminated cutaneous hyalohyphomycosis caused by Fusarium proliferatum in a patient with aplastic anemia. The Journal of dermatology, 50(6), e183–e184. https://doi.org/10.1111/1346-8138.16708), and Fusarium subglutinans (Chen, Q. X., Li, C. X., Huang, W. M., Shi, J. Q., & Li, S. F. (2010). Cutaneous hyalohyphomycosis caused by Fusarium subglutinans. European journal of dermatology : EJD, 20(4), 526–527. https://doi.org/10.1684/ejd.2010.0982). A relevant publication with great impact on the scientific community could offer in this section more insights (on Fusarium solani and oxysporum): Nucci, M., & Anaissie, E. (2002). Cutaneous infection by Fusarium species in healthy and immunocompromised hosts: implications for diagnosis and management. Clinical infectious diseases : an official publication of the Infectious Diseases Society of America, 35(8), 909–920. https://doi.org/10.1086/342328 . // Scedosporium apiospermum needs to be highlighted in the corresponding section. A representative publication is: Makino, K., Fukushima, S., Maruo, K., Egawa, K., Nishimoto, K., & Ihn, H. (2011). Cutaneous hyalohyphomycosis by Scedosporium apiospermum in an immunocompromised patient. Mycoses, 54(3), 259–261. https://doi.org/10.1111/j.1439-0507.2009.01797.x // Acremonium strictum is also a causative agent of cutaneous hyalohyphomycosis (though rare): Sharma, A., Hazarika, N. K., Barua, P., Shivaprakash, M. R., & Chakrabarti, A. (2013). Acremonium strictum: Report of a Rare Emerging Agent of Cutaneous Hyalohyphomycosis with Review of Literatures. Mycopathologia, Advance online publication. https://doi.org/10.1007/s11046-013-9709-1 // There is a reference in this Table, but in general, citations are lacking here. It is understandable not to cite publications for a Figure (when more details can be extracted from text), but you need to add more citations (even at the end of Table 1 legend altogether). The reference [40] in the Table does not agree the number of citations in the text.
- The mentioned typos are corrected. The suggested references are added. Additional references are added to meet the need for more references. We agree that table lacked the references – thank you for the input!
R1: Lines 132-135: Please rephrase the sentence, as it is difficult to read.
- The sentence is rephrased to: Despite previously considered an extremely rare pathogen in humans, it has the ability of causing infection of the skin and other sites in both immunosuppressed and healthy individuals. Hope now it is more understandable.
R1: Line 142: CVC probably refers to central venous catheters, but it has to be explained in the text as has not been mentioned before.
- Meaning of the acronym is given!
R1: Line 155: CNS acronym is first used in Table 1. Please make the necessary corrections.
- The acronym is used instead of the full name!
R1: Figure 4: Please correct the typo in “cellulitis”
- The typo is corrected!
R1: Line 240: In the section “Treatment” a significant publication is missing, which includes a summary of recommendations for the treatment of Purpureocillium lilacinum infections: Tortorano, A. M., Richardson, M., Roilides, E., van Diepeningen, A., Caira, M., Munoz, P., Johnson, E., Meletiadis, J., Pana, Z. D., Lackner, M., Verweij, P., Freiberger, T., Cornely, O. A., Arikan-Akdagli, S., Dannaoui, E., Groll, A. H., Lagrou, K., Chakrabarti, A., Lanternier, F., Pagano, L., … European Confederation of Medical Mycology (2014). ESCMID and ECMM joint guidelines on diagnosis and management of hyalohyphomycosis: Fusarium spp., Scedosporium spp. and others. Clinical microbiology and infection : the official publication of the European Society of Clinical Microbiology and Infectious Diseases, 20 Suppl 3, 27–46. https://doi.org/10.1111/1469-0691.12465
- We included the aforementioned significant publication and added the corresponding text under the section. Since the paper stated that optimal antifungal treatment for P.lilacinum has not yet been established, we kept information on sensitivity to antifungal agents originating from other papers.
R1: Line 272: An increased prevalence of invasive fusariosis has been noted in children with hematological malignancies who were treated with therapies that target cell surface antigens: Kyriakidis, I., Vasileiou, E., Rossig, C., Roilides, E., Groll, A. H., & Tragiannidis, A. (2021). Invasive Fungal Diseases in Children with Hematological Malignancies Treated with Therapies That Target Cell Surface Antigens: Monoclonal Antibodies, Immune Checkpoint Inhibitors and CAR T-Cell Therapies. Journal of fungi (Basel, Switzerland), 7(3), 186. https://doi.org/10.3390/jof7030186 .
- The aforementioned information with a corresponding reference has been added. Thank you!
R1: Line 552: Typo of “micology”
- Corrected!
R1: Line 555: Immunosuppression raises the suspicion of fungal involvement and it should be stated clearly in the differential diagnosis section. In one sentence, please refer to the reason for the failure of fungal prophylaxis in these cases. Moreover, host defenses against fungi rely on the sophisticated interplay between mucocutaneous barrier integrity (among others).
- In the very beginning of the section we included this information and referred on the reason for failure of fungal prophylaxis.
R1: General comment: The separate sections for each hyalohyphomycosis agent can be shortened, as the text is wordy and complicated to read. You should focus only on cutaneous involvement and cite treatment guidelines from well-respected sources and committees (and not case reports).
- Thank you for your kind input! We rearranged the text in sections 4-9 to mirror the structure of section 3. Now the manuscript is easier to read. Although your opinion on length of the sections is understandable, we did not shorten the sections for each of the agents, since some basic information about the fungi is necessary – this amount of information on each of the fungi can be found in similar papers, including the paper you asked us to include (ESCMID and ECMM joint guidelines on diagnosis and management of hyalohyphomycosis). We did put focus on cutaneous lesions, as it can be seen through the text and the subtitles, but the complete segregation of the cutaneous infections from all other forms of infections was not our intention, nor is possible, especially because cutaneous infections can progress into invasive and disseminated infections and disseminated infection usually have some form of cutaneous involvement. Furthermore, information on pathogen and the spectrum of infections it can cause is needed for better understanding of cutaneous infections. Also, we understand the notion that we cited too much case reports, but this does not come as surprise, since most of the fungi of this group are rare cause of disease in humans. We tried to collect recent and relevant case reports to form a mosaic view of cutaneous infections caused by rare fungi. But we also included detailed and respectful review articles on the topic, such as one from Sprute et al. and Sal et al.
R2: 1. Enhancement of Table 1 by expanding it to include an additional column. This column should detail the level of antifungal resistance observed, list the specific antifungal agents that have shown reduced efficacy, and identify those that remain effective in treating the infection.
- Although it was not our initial intention to detail the antifungal resistance, we agree on your suggestion that the table could benefit from this expansion, especially because this is one of the shared characteristics of the fungi belonging to this group. Therefore, we added an additional column containing information about gross sensitivity to most used antifungal agents, coupled with corresponding references. Thank you for the idea!
R2: 2.The sections covering other fungal groups (i.e., sections 4 to 9) could benefit from expansion and reorganization to mirror the structure of section 3 (i.e., Purpureocillium lilacinum). This would involve organizing each section into distinct subsections focusing on epidemiology, pathogenesis, cutaneous infections, diagnosis, and treatment, similar to the format used in section 3.
- We absolutely agree on this comment! We reorganized the sections 4 to 9 to mirror the structure of the section 3. Now the manuscript is easier to read and the information are more accessible. We did partial expansion to some parts of the text to make it more understandable. Thank you for this great idea!
- R2: There are multiple grammatical and typographical errors that would need to be corrected. Some of them are detailed below:
- Potentially wrong word choice in line 98 ‘cynically presented as…’
- Potentially wrong word choice in line 100 ‘…with each of the efflorescence being associated…’
iii. Change in line 104 ‘timely suspect cutaneous…’ to ‘timely diagnose cutaneous…’
- Correct line 145 ‘Also, it is also opportunistic…’, remove redundant ‘Also’
- Change in line 156 ‘…and consequential dissemination...’ to ‘…and consequent dissemination...’
- We did all the mentioned corrections as it can be seen in the text. Thank you for your input!
Reviewer 2 Report
Comments and Suggestions for Authors
The authors in this review present a comprehensive review of cutaneous hyalohyphomycosis infections caused by P.lilacinum and other fungi within the group including genera of Fusarium, Penicilium, Scedosporium, Scopulariopsis, Acremonium, and Trichoderma.
This is a well-structured detailed review that can be accepted pending the changes listed below
1. Enhancement of Table 1 by expanding it to include an additional column. This column should detail the level of antifungal resistance observed, list the specific antifungal agents that have shown reduced efficacy, and identify those that remain effective in treating the infection.
2. The sections covering other fungal groups (i.e., sections 4 to 9) could benefit from expansion and reorganization to mirror the structure of section 3 (i.e., Purpureocillium lilacinum). This would involve organizing each section into distinct subsections focusing on epidemiology, pathogenesis, cutaneous infections, diagnosis, and treatment, similar to the format used in section 3.
3. There are multiple grammatical and typographical errors that would need to be corrected. Some of them are detailed below:
i. Potentially wrong word choice in line 98 ‘cynically presented as…’
ii. Potentially wrong word choice in line 100 ‘…with each of the efflorescence being associated…’
iii. Change in line 104 ‘timely suspect cutaneous…’ to ‘timely diagnose cutaneous…’
iv. Correct line 145 ‘Also, it is also opportunistic…’, remove redundant ‘Also’
v. Change in line 156 ‘…and consequential dissemination...’ to ‘…and consequent dissemination...’
Comments on the Quality of English LanguageThere are multiple grammatical and typographical errors that would need to be corrected. Some of them are detailed below:
i. Potentially wrong word choice in line 98 ‘cynically presented as…’
ii. Potentially wrong word choice in line 100 ‘…with each of the efflorescence being associated…’
iii. Change in line 104 ‘timely suspect cutaneous…’ to ‘timely diagnose cutaneous…’
iv. Correct line 145 ‘Also, it is also opportunistic…’, remove redundant ‘Also’
v. Change in line 156 ‘…and consequential dissemination...’ to ‘…and consequent dissemination...’
Author Response

(The authors gave the same response as above.)
